# A Novel Homozygous Nonsense Variant in the *DYM* Underlies Dyggve-Melchior-Clausen Syndrome in Large Consanguineous Family

**DOI:** 10.3390/genes14020510

**Published:** 2023-02-17

**Authors:** Abu Bakar, Sulaiman Shams, Nousheen Bibi, Asmat Ullah, Wasim Ahmad, Musharraf Jelani, Osama Yousef Muthaffar, Angham Abdulrhman Abdulkareem, Turki S. Abujamel, Absarul Haque, Muhammad Imran Naseer, Bushra Khan

**Affiliations:** 1Department of Biochemistry, Abdul Wali Khan University Mardan, Mardan 23200, Pakistan; 2Department of Bioinformatics, Shaheed Benazir Bhutto Women University, Peshawar 25120, Pakistan; 3Novo Nordisk Foundation Center for Basic Metabolic Research, Faculty of Health and Medical Sciences, University of Copenhagen, 2100 Copenhagen, Denmark; 4Department of Biochemistry, Quaid-I-Azam University, Islamabad 45320, Pakistan; 5Rare Diseases Genetics and Genomics, Centre for Omic Sciences, Islamia College Peshawar, Peshawar 25120, Pakistan; 6Department of Pediatrics, Faculty of Medicine, King Abdulaziz University, Jeddah 21589, Saudi Arabia; 7Faculty of Science, Department of Biochemistry, King Abdulaziz University, Jeddah 21589, Saudi Arabia; 8Center of Excellence in Genomic Medicine Research, King Abdulaziz University, Jeddah 21589, Saudi Arabia; 9Department of Medical Laboratory Technology, Faculty of Applied Medical Sciences, King Abdulaziz University, Jeddah 21589, Saudi Arabia; 10Vaccines and Immunotherapy Unit, King Fahd Medical Research Center, King Abdulaziz University, Jeddah 21589, Saudi Arabia; 11King Fahd Medical Research Center, King Abdulaziz University, Jeddah 21859, Saudi Arabia

**Keywords:** *DYM* gene, Dyggve-Melchior-Clausen Syndrome, homozgosity mapping, sanger sequencing, novel homozygous, non-sense variant

## Abstract

(1) Background: Dyggve-Melchior-Clausen Syndrome is a skeletal dysplasia caused by a defect in the *DYM* gene (OMIM number 607461). Pathogenic variants in the gene have been reported to cause Dyggve-Melchior-Clausen (DMC; OMIM 223800) dysplasia and Smith-McCort (SMC; OMIM 607326) dysplasia. (2) Methods: In the present study, large consanguineous families with five affected individuals with osteochondrodysplasia phenotypes were recruited. The family members were analyzed by polymerase chain reaction for homozygosity mapping using highly polymorphic microsatellite markers. Subsequent to linkage analysis, the coding exons and exon intron border of the *DYM* gene were amplified. The amplified products were then sent for Sanger sequencing. The structural effect of the pathogenic variant was analyzed by different bioinformatics tools. (3) Results: Homozygosity mapping revealed a 9 Mb homozygous region on chromosome 18q21.1 harboring *DYM* shared by all available affected individuals. Sanger sequencing of the coding exons and exon intron borders of the *DYM* gene revealed a novel homozygous nonsense variant [DYM (NM_017653.6):c.1205T>A, p.(Leu402Ter)] in affected individuals. All the available unaffected individuals were either heterozygous or wild type for the identified variant. The identified mutation results in loss of protein stability and weekend interactions with other proteins making them pathogenic (4) Conclusions: This is the second nonsense mutation reported in a Pakistani population causing DMC. The study presented would be helpful in prenatal screening, genetic counseling, and carrier testing of other members in the Pakistani community.

## 1. Introduction

Skeletal dysplasias (SDs), for the most part named as congenital bone disorders, are inherited bone and cartilage defects that impair the stability, development, and morphology of the bones and cartilage [1,2]. There are 436 reported skeletal disorders, which are divided in 42 different groups based on their molecular, clinical, radiographic, and biochemical properties [3,4]. Hundreds of genes have been reported to carry pathogenic mutations causing these skeletal disorders [2]. Skeletal deformities are diversified into two main groups based on mutations in extracellular matrix proteins, ion channels, cellular transporters, transcription factors, signaling molecules, and RNA processing machinery. These two groups are the osteochondrodysplasias and dysostoses. Dysostoses occur mostly from mutations in the patterning genes, thus characterized by abnormalities in a single bone or a group of bones. Inadequate growth and development of the bones and cartilage is termed as osteochondrodysplasia [5].

Dyggve-Melchior-Clausen (DMC) syndrome is an autosomal recessive skeletal disorder, which is very rare as illustrated by an aberrant ossification of the epiphyses and metaphysis of long bone, as well as deformity of the vertebrae associated with intellectual dysfunction. Smith-McCort (SMC) is a variant of DMC with normal intelligence. SMC and DMC syndromes come under the category of autosomal recessive spondylo-epi-metaphyseal dysplasia (SEMD), which is a class of osteochondrodysplasia induced by defects in the *Dymeclin* (*DYM*) gene located at chromosome 18q21.1 [6,7,8]. Short trunk and limbs, barrel chest, brachydactyly, varus (bowlegs) and valgus (knock knee) malformation of the knees, facial dysmorphism, limited joint movement, and micropenis are primary phenotypes of the DMC syndrome. Radiological characteristics comprise of hypoplastic odontoid process, double humped vertebrae, platyspondyly, outwards and sideways curvature of the spine, epiphysis, metaphysis, aberrant ossification of long bones, lacy iliac crests, and anterior beaking of vertebral bodies. While the skeletal abnormalities seen in patients having DMC syndrome are comparable to those seen in Smith-McCort dysplasia, only the former has been linked to intellectual dysfunction [7,9]. In both disorders, the *DYM* mechanism is disrupted, resulting in pathological cartilage histology with distorted chondrocyte columns containing rough endoplasmic reticulum inclusions, degenerating cells in the growth plate [10,11,12,13]. It is distinguished from Morquio’s disease (Mucopolysaccharidoses type-IV, MIM 253010) by the absence of additional clinical symptoms, such as corneal opacity and keratan sulphate excretion in urine, as well as the existence of certain radiological abnormalities [14].

The *DYM* gene encoding a polypeptide of 669 amino acids is composed of six transmembrane regions, with the N-terminus situated inside the cytoplasm. Its occurrence is common in normal cells but is most abundant in chondrocytes, osteoblasts, and fetal brain cells [6]. It controls secretory pathways associated with the Golgi that are necessary in endochondral bone formation in early development [15]. A complete lack of the *DYM* protein in DMC patient could indicate brain damage. Dupuis et al. (2013) discovered that depletion of the *DYM* protein causes abnormalities in brain development and postnatal onset microcephaly in *DYM*^−^/^−^ knockout mice [16].

In the present study, a consanguineous family of Pakistani origin with five patients presenting osteochondrodysplasia related features was clinically and genetically investigated.

## 2. Materials and Methods

### 2.1. Ethical Considerations and Family History

The Institutional Review Board (IRB) of Abdul Wali Khan University, Mardan, Pakistan gave its approval to the current study. The family was founded from the Khyber Pakhtunkhwa province’s Swat area. Drawing the pedigree was helped by family elders. All participating family members were asked to sign a written informed consent form that was authorized by the IRB in order to conduct the study and report the results in accordance with the Helsinki declaration, which also included participant pictures and radiographs. The autosomal recessive nature of the disorder’s inheritance was clearly demonstrated through pedigree testing.

A medical officer evaluated five afflicted family members (III-1, III-2, III-3, IV-1, and IV-2) at the local government hospital (Figure 1A). The patient underwent axial and appendicular skeletal X-rays at the same facility (Figure 1B). A nonelastic plastic tape was used to measure the participant’s height (in cm) while they were standing straight and without shoes. The patients’ weights (kg), measured to the closest 0.1 kg on a digital scale, were taken without shoes and wearing light clothing.

### 2.2. Collection of Blood Samples and Extraction of Genomic DNA

Eleven volunteers submitted blood samples for the study, comprising six normal people (II-1, II-2, III-4, III-5, III-6, and IV-3) and five impacted patients (III-1, III-2, III-3, IV-1, and IV-2). Using the phenol/chloroform method, genomic DNA was extracted from the blood samples and measured with a Nanodrop-1000 spectrophotometer (Thermal Scientific, Wilmington, MA, USA). For use in the PCR amplification, the genomic DNA was diluted to40 ng/L.

### 2.3. Genotyping and Homozygosity Mapping

Due to the pedigree’s consanguinity and the disorder’s autosomal recessive inheritance pattern, homozygosity mapping using polymorphic microsatellite markers was carried out using DNA from both affected and unaffected family members. The markers were chosen from genomic areas previously described for their correlation with Dyggve-Melchior-Clausen and other related disorders (Acromesomelic dysplasia Maroteaux type and Morquio syndrome A). For genotyping, at least eight highly polymorphic microsatellite markers that are each adjacent to the *DYM*, *NPR2*, and *GALNS* genes were chosen.

Polymerase chain reaction (PCR) was carried out according to the established protocol to amplify the markers. The amplified products were analyzed using 8% nondenaturing polyacrylamide gels stained with ethidium bromide, and the genotypes were determined by eye inspection. In afflicted patients, an area of homozygosity was found using haplotype analysis (Figure 2A).

### 2.4. Sequencing DYM

The *DYM* was bidirectionally sequenced utilizing the Sanger di deoxy chain termination technique once the family’s linkage was established. *DYM’s* 16 coding exons primer pairs were constructed using Primer3Plus software (available at http://www.bioinformatics.nl/cgi-bin/primer3plus/primer3plus.cgi, accessed on 10 April 2022). With the designed primers (Appendix A) and under standard protocols, DNA from five affected patients (III-1, III-2, III-3, IV-1, and IV-2) and six healthy people (II-1, II-2, III-4, III-5, III-6, IV-3) were PCR amplified and sequenced. The Big Dye Terminator v3.1 Cycle Sequencing Kit (Life Technologies, Carlsbad, CA, USA)was used to sequence the PCR-amplified products after they had been purified using a commercial kit (Axygen, CA, USA). Using the readily available online datasets, such as 1000 Genomes, Exome Variation Server (EVS), gnomAD (V.2.1.1, V.3.1.2, and ExAC), as well as 135 in-house exomes, the variant frequency in the general population was examined.

### 2.5. In Silico Analysis

Pathogenicity of the identified variant was analyzed using MutationTaster (https://www.mutationtaster.org/, accessed on 15 August 2022), BayesDel (https://fengbj-laboratory.org/cgi-sys/suspendedpage.cgi, accessed on 15 August 2022), CADD (https://cadd.gs.washington.edu/snvDANN, accessed on 15 August 2022) (https://cbcl.ics.uci.edu/public_data/DANN/, accessed on 15 August 2022) VarSome (https://varsome.com/, accessed on 15 August 2022), PROVEAN (http://provean.jcvi.org/index.php, accessed on 15 August 2022), and SIFT (https://sift.bii.a-star.edu.sg/, accessed on 15 August 2022). ACMG guidelines were followed to classify the pathogenicity of the identified variant (Appendix A).

### 2.6. Secondary and Tertiary Structure of DYM

Crystal structure of *DYM* protein is not available in the protein data bank (PDB) (https://www.rcsb.org/, accessed on 20 August 2022); therefore, the tertiary structure of *DYM* was predicted through I-Tasser server [17] and validated through Ramachandran plot (https://zlab.umassmed.edu/bu/rama/, accessed on 20 August 2022) for stereochemistry analysis. Chimera 1.5.6 tool was used for the visualization and energy minimization of predicted model. The predicted model was subjected to structural analysis in the PDBsum [18] in order to get the full annotation of the predicted model.

### 2.7. Hydropathy and Conservation Analysis

Membrane Protein Explorer (MPEx) tool was used to predict the transmembrane region of the *DYM* protein on the basis of the experimental hydrophobicity value of amino acids [19]. Evolutionary conservation of Leu402 in the *DYM* protein was evaluated through multiple sequence alignment across seven species performed in clustalW server (https://www.genome.jp/tools-bin/clustalw, accessed on 22 August 2022).

### 2.8. DYM Protein Interaction Network

STRING and IntAct (https://www.ebi.ac.uk/intact/home, accessed on 24 August 2022) databases were used to retrieve the functional protein interaction network of the *DYM* protein. STRING predicts different types of interactions and award scores to each interaction from 0–1 (lower to higher). IntAct is a molecular interaction database where all the interactions used are submitted or the literature is curated [20].

## 3. Results

### 3.1. Clinical Description

At the time of study, age of the affected patients ranged from 10 to 40 years. Clinical phenotypes of the affected patients IV-1 and IV-2 (Figure 1A(a,b)),showed features of DMC, a rare variety of SEMD, which often resembles Morquio syndrome. Phenotypes that were noticed in affected patients were short trunk and limbs, short humerus, abnormal walking due to varus and valgus deformation within the knee, pigeon chest, limited joint mobilization, and variable intellectual disability (Table 1). Extreme shyness and fear in some patients (IV-1 and IV-2) and variable speech defects were also observed in patients (III-2, III-3, IV-1 and IV-2) with the above clinical symptoms. Patient III-2 suffered from a subtype of articulation disorder of speech impairment i.e., apraxia of sound where the individual is unable to form words due to intellectual disability, while patients III-3, IV-1, and IV-2 experienced dysarthria, a condition where patients slur their words because of intellectual disability. Patient III-1 had normal speech sound. All the affected members of the family showed autism. Radiographic examination of all the afflicted ones displayed numerous anomalies comprising flattened vertebrae alongside diminished vertical length, outwards and sideways curvature of the spine, anterior beaking of vertebral bodies, lacy iliac wings, hypoplastic odontoid, and laterally dislocated irregularly ossified femoral heads (Figure 1B). Radiographic presentation of the thoracic, pubic, knee joints, arm of patient (IV-1) (Figure 1B(a–d)), and thoracic, pubic, and knee joints of the patient (IV-2) (Figure 1B(e,f)) revealed clinical phenotypes of DMC syndrome. Affected individual IV-1 also showed a bowed humerus (Figure 1B(d)).

### 3.2. Homozygosity and Variant Identification

Based on the clinical characteristics of the afflicted members of this family, linkage analysis was performed with microsatellite markers flanking *NPR2, GALNS,* and *DYM*. Among these markers, D18S1094, D18S1143, D18S1118, D18S467, D18S1126, D18S845, D18S846, and D18S539,closely located to the region harboring *DYM* (18q21.1), were shown to be heterozygous in unaffected family members and homozygous in the afflicted cases (Figure 2A).

Sanger sequencing discovered a novel homozygous nonsense disease causing the variant [c.1205T>A, p.(Leu402*)] in exon 11 of the *DYM* gene that segregated the disease phenotype within the family (Figure 2C). In several datasets, including EVC, 1000 genome, gnomAD, and 135 Pakistani in-house exomes of controlled individuals, the mutation was not found in a homozygous condition. Eight out of nine various human mutation effect prediction software tools predicted the identified nonsense variant [c.1205T>A, p.(Leu402*)] in the *DYM* to be damaging/deleterious. According to ACMG classification, the variant p.(Leu402*) in *DYM* was classified pathogenic (Appendix A). The variant’s LOVD submission has been made with ID # 000087156.

### 3.3. Secondary and Tertiary Structure of DYM protein

The secondary and tertiary structure of the *DYM* protein was predicted to get insight into the structural effects of the mutation. The identified mutation p.(Leu402*) was located in the α helix in the intracellular domain of the *DYM* protein (Figure 3a). The tertiary structure of *DYM* was predicted through I-Tasser server with C-score and Tm-score of 0.8 and 0.7, respectively. The validity and stereochemistry of the *DYM* predicted model was evaluated through Ramachandran plot indicating that 98% of residues were in the favored conformation. The domain architecture of the protein showed the extracellular domain, intracellular domain, and transmembrane domain (Figure 3b,c).

### 3.4. Hydropathy and Conservation Analysis

Membrane Protein Explorer (MPEx) tool was used for the analysis of hydrophobicity of individual amino acids [19]. The transmembrane region was predicted using an experimental value of hydrophobicity based on biological and physical analysis. This tool is used to evaluate the effect of a particular mutation on the transmembrane region. Leu402 is located in the extracellular domain. The identified mutation in the present study resulted in truncated *DYM* protein with loss of TM and intra cellular domain (Figure 4A(a)). Functional importance of the amino acid is evaluated through evolutionary conservation analysis. Evolutionary conservation of Leu402 was validated through multiple sequence alignment of the *DYM* protein across seven species. Results indicated conservation of Leu402 across seven homologs from human to zebrafish (Figure 4A(b)).

### 3.5. Effect of Variant Leu402* on Protein Function and Interactions

The identified mutation p.(Leu402*) mapped in the extracellular domain of the *DYM* protein resulted in loss of the TM and intracellular domain (Figure 4B(a,b)), and consequently the interactions of *DYM* with other interacting partners will be diminished (Figure 4B(c)). The interaction of the protein is a measure of its functional stability. The truncated protein with loss of binding interactions resulted in disease phenotype.

## 4. Discussion

The current study portrays a large consanguineous family, of Pakistani origin, with segregating DMC syndrome in an autosomal recessive manner. DMC is featured by various skeletal phenotypes i.e., short stature and short trunk associated with intellectual disability. Coarse facial features, rhizomelia, microcephaly, and bone malformation are also the indications of DMC [16,32]. The majority of the DMC related characteristics discovered in our affected members were identical to those previously reported in similar cases [22,30]. The affected members of our family depicted a rare feature of DMC i.e., autism, as reported in other cases [26,27]. The presence of mild to severe intellectual disability in all available five affected family members led us to classify them as DMC patients rather than SMC patients.

Along with the typical dysmorphic DMC signs, our patients presented with variable speech defects, observed in patients III-2, III-3, IV-1, and IV-2. Four of the affected members of the family suffered from articulation disorders of speech impairment, such as apraxia of sound and dysarthria in patient III-2 and patient III-2, IV-1, and IV-2, respectively. Bowed humerus in individual IV-1 (Figure 1B(d)) and short length humerus(Figure 1A(b)) of affected individual IV-2 were also noted. One of them was also found to be overweight (III-2). To the best of our knowledge, these additional phenotypic features were not reported earlier. The intra and inter familial phenotypic variations may be due to the effects of potential modifier genes and the different ages of the patients.

DNA sequencing discovered a novel homozygous nonsense disease causing variant p.(Leu402*) in exon 11 of the *DYM* gene in our patients. The variant involved the transition of T at nucleotide position 1205 to A replacing leucine (TTG) by a stop codon TAG (Figure 2B). The protein is prematurely terminated by this novel variant at amino acid position 402. Therefore, it is expected that the discovered variation would result in loss of function by triggering nonsense-mediated decay or production of an aberrant peptide. Similarly, most of the previously identified *DYM* variants, linked to DMC, caused frameshifts by introducing a premature stop codon either by nucleotide substitutions or duplications, deletions, or insertions [6,7,8,10,21,22,23,24,25,28,29,30,31,33]. These variants may lead to the loss of dymeclin function (Appendix A). According to current and other reported cases, patients with DMC have an overall good health and often live into adulthood. However, with time these individuals develop dislocations of the atlanto-axial and hip joints, which can lower their quality of life [21,27,30,33]. Similarly, the patient in our family was also overweight; therefore, it is highly likely they may experience some other comorbidity.

The *DYM* gene produces a polypeptide having 669 amino acids that is comprised of six transmembrane segments, with the N-terminus in the cytoplasm. It is found in all normal cells but is most abundant in osteoblasts, chondrocytes, and fetal brain cells [6]. During early development of endochondral bone formation, *DYM* takes an essential part in the regulation of the Golgi associated secretory pathway. The gene also participates in the early stages of brain development [34]. DMC dysplasia and Smith-McCort (SMC) dysplasia have both been linked to variations in this gene. The majority of the traits are shared by the two disorders, as well as two pathognomonic signs of lacy iliac crest and bi humped vertebral bodies. Intellectual disability that is linked with DMC, but not with SMC dysplasia, is the distinguishing trait between the two syndromes. DMC is caused by protein truncating sequence variants in *DYM*, whereas SMC is caused by missense variants in the same gene [22]. As a result, the current study’s clinical findings were backed up by genetic findings on the basis of a previous link of nonsense mutations in the gene with DMC. So far, HGMD Professional 2018.4 (http://www.hgmd.cf.ac.uk/ac/gene.php?gene=DYM/, accessed on 30 August 2022) has a registry of 44 pathogenic variants (missense/non-sense (18 variants), splicing substitutions (11 variants), deletions (11 variants), and insertions/duplications (4 variants)) in *DYM*, causing DMC, SMC1, intellectual disability, and other dysmorphic phenotypes (Appendix A). Only twenty-nine *DYM* variants have been reported to cause DMC (HGMD Professional 2018.4) in patients from various countries around the globe, such as Tunisia, Morocco, Lebanon, Japan, Spain, India, and Pakistan [6,25].

In the current study, we have reported the 2ndhomozygous nonsense p.(Leu402*) and fifth loss of function variant in *DYM* in a consanguineous family of Pakistani origin. The other four DMC syndrome causing mutations were single base pair homozygous insertion/deletions and substitution (96insT; 763delA; 1004delT; and c.59T>A) (7,8,16,24). Due to high rate of consanguinity in the Pakistani population, the number of autosomal recessive disorders is increasing day by day. Therefore, it is necessary to conduct genetic screening of affected patients and perform carrier testing and genetic counselling of the affected families to prevent the disease.

## 5. Conclusions

In a nutshell, we have characterized an extensive consanguineous family of Pakistani descent segregating the DMC that is linked to speech impairments in an autosomal recessive manner. Sequence analysis of the *DYM* gene in the Pakistani population revealed a 2^nd^ novel homozygous nonsense and a fifth loss-of-function, disease-causing variant p.(Leu402*). The research not only broadened the gene’s mutational spectrum, but it may also help in the prevention of disease through prenatal screening, carrier testing, and genetic counseling.

## Figures and Tables

**Figure 1 genes-14-00510-f001:**
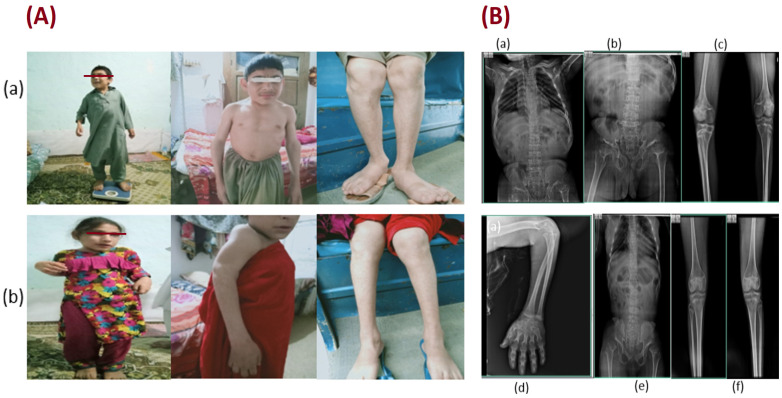
Clinical phenotypes of affected individuals of the family: (**Panel A**): Phenotypes of the affected individuals (IV-1 and IV-2) (**a**,**b**). Clinical features that were noticed in affected individuals were short trunk and limbs, short humerus, abnormal walking due to varus and valgus deformation within the knee, pigeon chest, limited joint mobilization, and variable mental retardation. (**Panel B**): Radiographic examination of the afflicted, who display numerous anomalies comprising flattened vertebrae alongside diminish vertical length, outwards and sideways curvature of the spine, anterior beaking of vertebral bodies, lacy iliac wings, hypoplastic odontoid, and laterally dislocated irregularly ossified femoral heads. Radiographic presentation of the thoracic, pubic, knee joints, arm of patient (IV-1) (**a**–**d**) and thoracic, pubic, and knee joints of the patient (IV-2) (**e**,**f**) reveal clinical phenotypes of DMC syndrome.

**Figure 2 genes-14-00510-f002:**
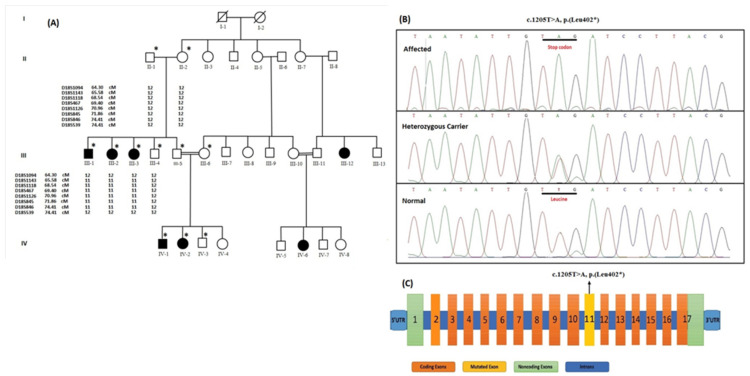
Haplotypes and sequence analysis of the family: (**Panel A**): Pedigree of consanguineous family segregating DMC in autosomal recessive manner. Affected and unaffected individuals are represented by filled and empty symbols, respectively. An asterisk (*) sign over the symbol shows the individuals from which blood is drawn. A consanguineous union is represented by a double line. On the symbols, a cross line denotes a deceased person. Each symbol is followed by the haplotypes of the closely related microsatellite markers on chromosome 18q21.1, with haplotypes 11 and 12 denoting homozygous and heterozygous individuals, respectively. (**Panel B**): Chromatogram showing sequence analysis of the *DYM* gene. Upper panel shows affected member, middle indicates heterozygous carrier state, while the lower one indicates wild-type individual. Mutation involved transition of T to A replacing Leucine by a stop codon TAG at 402 position. Asterisk (*) shows stop codon. (**Panel C**): Schematic representation of all the 17 exons of the *DYM* gene. Orange color indicates coding exons (2–17), pale green color depicts noncoding exon (1), while yellow color of exon 11 shows point of mutationc.1205T>A, p.(Leu402*). Asterisk (*) shows stop codon.

**Figure 3 genes-14-00510-f003:**
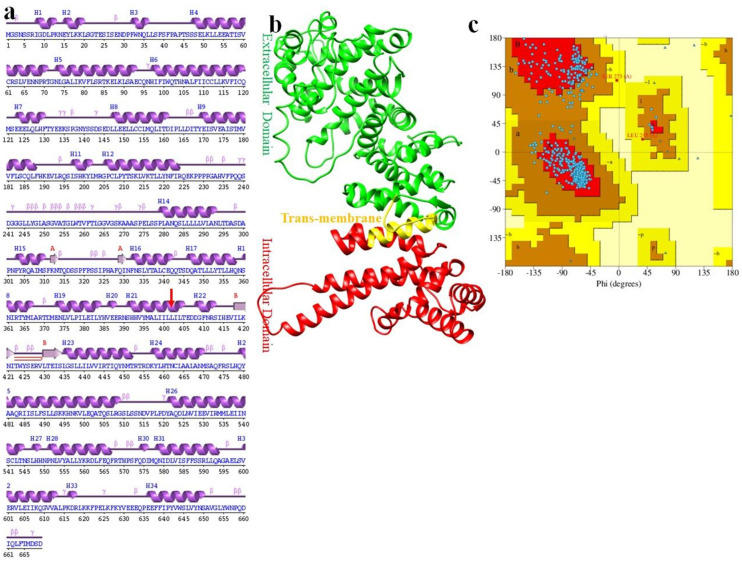
Secondary and tertiary structure of *DYM* protein. (**a**) Secondary structure of *DYM* with mutation position shown in red arrow. Sheets and helices are shown in purple color and βturns are shown in red. (**b**) Tertiary structure of *DYM* in ribbon representation. Different domains are shown in their respective colors.(**c**) Ramachandran plot of *DYM* tertiary structure.

**Figure 4 genes-14-00510-f004:**
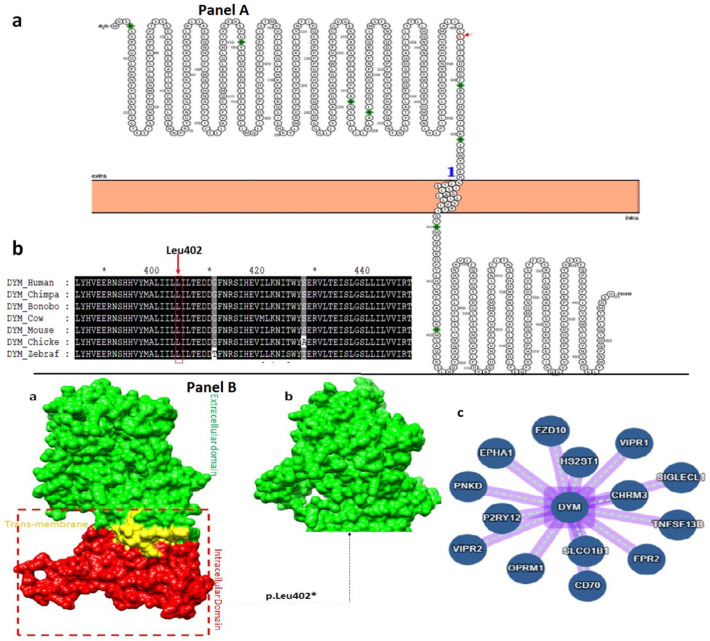
Hydropathy and evolutionary conservation of the *DYM* protein and hydrophobic representation of wild-type and mutant *DYM* protein model. (**Panel A**): (a) Hydropathy plot; (b) evolutionary conservation across seven species. (**Panel B**): (a) Represents the wild-type protein; (b) represents the mutant protein with truncated domain; and (c) represents the interaction network of *DYM*, while nodes represent the interacting proteins and edges represent the interaction.

**Table 1 genes-14-00510-t001:** *DYM* mutations causing DMC and SMC in different ethnic groups with correlation of clinical phenotypes of our family and previously reported cases.

Nucleotide Change/ Amino Acid Change	Ethnic Origin	Skeletal Disorders (Short Trunk, Double Humped Vertebrae, Lacy Iliac Crest, Brachydactyly, Barrel Chest, and Kyphoscoliosis)	Microcephaly	Autistic Features	Rectal Prolapse	Waddling Gait	Ectodermal Features	Micropenis	Vision Problem	Atlantoaxial Instability	References
**Dyggve-Melchior-Clausen Syndrome**											
48C>G/Y16X	Dominia	+	−	−	−	−	−	−	−	−	[7]
59T>A/L20X	Pakistan	+	−	−	−	+	+	−	−	−	[8]
Duplication exon 2/ S47Rfs	Lebanon	+	−	−	−	−	−	−	−	−	[21]
95_96insT/W33Lfs*14	Pakistan	+	+	+	−	+	+	−	−	−	[22]
208C>T/R70X	Tamil, India	+	−	−	−	−	−	−	−	−	[21]
369T>A/1405A>T/ Y132X/N469Y	Not stated	+	−	−	−	−	−	−	−	−	[7]
580C>T/R194X	Tunisia, Turkey	+	+		+	−	−	−	−	−	[6,23]
610C>T/R204X	Morocco	+	+		+	−	−	−	−	−	[6,23]
656T>G/1877delA/ L219X/K626Nfs*92	Morocco	+	+		+	−	−	−	−	−	[6,23]
763delA/T254Qfs*9	Pakistan	+	−	−	−	−	−	−	−	−	[7]
1028_1056del29/ Q343Lfs*8	Turkey	+	+	+	+	+	+	−	+	−	[24]
1172_1173insC/ H392Tfs*17	India	+	+	+	−	−	−	−	−	+	[25]
1670delT/L557Rfs*20	Japan	+	+	+	+	+	−	−	−	−	[26]
**1205T>A/L402X**	**Pakistan**	**+**	**−**	**+**	**−**	**+**	**+**	**−**	**−**	**−**	**Our case**
1363C>T/R455X	Tamil, India	+	−	−	−	−	−	−	−	−	[21]
1447C>T/Q483X	Morocco	+	+	+	−	−	−	−	−	−	[6,27]
G>C 34 bp 3′ of exon 1	Portugal	+	+	+	−	−	−	−	−	−	[27]
1877delA/K626Nfs*92	Morocco	+	+	−	+	−	−	−	−	−	[6,23]
1878delA/K626Nfs*94	Morocco	+	+	−	−	−	−	+	+	+	[10,28]
1938delTGTCT/ L646Lfs*61	Georgia	+	+	+	+	+	−	−	−	+	[29]
Repetition 4 copies exon 14/A525Ffs	Gujerati, India	+	−	−	−	−	−	−	−	−	[21]
IVS3 194-1G>A	Lebanon	+	+	+	+	+	−	−	−	+	[27,29]
IVS4 288-2A>G/ IVS7 621-2A>G	Spain	+	+	+	−	−	−	−	−	−	[27]
IVS5 422-2A>G/ IVS7 621-2A>G	Spain	+	−	−	−	−	−	−	−	−	[30]
IVS7 621-2A>G	Chile, Argentina	+	−	+	−	+	+	−	−	−	[30]
IVS10 1125+1G>T	Morocco	+	+		+	−	−	−	−	−	[6,23]
IVS11 1252-1G>A	Lebanon	+	+		+	−	−	−	−		[6,23]
IVS15 1746+3G>T	Indones	+	+	+		+	−	−	−	+	[31]
**Smith-McCort Syndrome**											
IVS7 621-2A>G/259G>A/ E87K	Gaum	+	−	−	−	−	−	−	−	−	[7]
1624T>C/C542R	Portugal	+	−	−	−	−	−	−	−	−	[9]

(*) means stop codon.

## Data Availability

All the relevant data is available within the manuscript and its Appendix A.

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
