# Peer review of "A Novel Homozygous Nonsense Variant in the DYM Underlies Dyggve-Melchior-Clausen Syndrome in Large Consanguineous Family"

_genes, 2023, doi:10.3390/genes14020510_

Round 1

Reviewer 1 Report

Summary: The author identified a novel ultrarare non-sense variant of DYM through genetic work up on a multi-generation family. In silico predication support a loss of function mechanism of the variant. The report is overall well written.

Major comments:

Skeletal deformities are prominent. However, the intellectual dysfunction of proband does not well support. In the text, the author use ‘brain damage’, is there any other supporting experiment/data, e.g., brain MRI, what’s the head circumference and z-score at the age of measurement?

The pedigree analysis suggests potential autosomal recessive inheritance and through the targeted sanger sequencing on related genes did find non-sense variant of DYM. Based on the pedigree and sanger data, Individual II-1 and II-2 does carrier the same pathogenic variant. Do they come from the similar genetic background e.g., same village or clan etc. Or does the variant locate at mutation hotspot?

The author explored potential product of a non-sense variant and predicated truncated protein product(Figure 4b). The author should comment on what the remaining function of the truncated protein which may related the phenotype variability observed of the case.

Minor review:

1.      Table 1 is hard to read with the current format. Consider reformatting or align the row correctly

2.      Line 200, use ‘SEMD’ as it has been defined in the introduction.

Author Response

Dear Reviewer,

Thank you for your time to critically review our manuscript and for your positive and valuable suggestions.

Major comments:

Comment 1: Skeletal deformities are prominent. However, the intellectual dysfunction of proband does not well support. In the text, the author use ‘brain damage’, is there any other supporting experiment/data, e.g., brain MRI, what’s the head circumference and z-score at the age of measurement?

Authors Response: Thanks for your comments. As the patients in the present study showed autistic features and speech impairment including apraxia and dysarthria. Previously, apraxia and dysarthria have been associated with brain damage in DYM-mutated patients; therefore, we concluded that our patients may also have brain damage leading to speech impairment. We requested parents of the patients for patients’ brain MRI to look at the brain structure, but they declined. Head circumference at the time of study was measured and found in normal range.

Comment 2: The pedigree analysis suggests potential autosomal recessive inheritance and through the targeted Sanger sequencing on related genes did find non-sense variant of DYM. Based on the pedigree and Sanger data, Individual II-1 and II-2 does carrier the same pathogenic variant. Do they come from the similar genetic background e.g., same village or clan etc. Or does the variant locate at mutation hotspot?

Authors Response: Thank you for the kind comments. The individuals II-1 and II-2 belong to same village and caste but they were not first-degree cousin. Therefore, we performed homozygosity mapping based on polymorphic microsatellite markers harboring DYM. The linkage revealed almost 10 MB region of homozygosity showing that the individuals come from same genetic background.

 Minor review:

Comment 1: Table 1 is hard to read with the current format. Consider reformatting or align the row correctly

Response: Corrected as per your kind suggestion.

 Comment 2:     Line 200, use ‘SEMD’ as it has been defined in the introduction.

Response: Corrected as per your kind suggestion.

Reviewer 2 Report

The paper is interesting and brings new information.

Review typing.

Figure 1: if possible, improve the photographs. C, D and E are uninformative.

Table 1: improve the table. Maybe just lower the font size.

The caption of Figure 2 was not clear to me. It gives the impression that there are individuals with the haplotype under investigation who are affected and who are not affected. As well as affected and unaffected individuals without the haplotype. I suggest making this information clearer. If there really are affected and non-affected carriers and non-carriers, I suggest exploring this further in the paper.

Author Response

Dear Reviewer,

Thank you for your time to critically review our manuscript and for your positive and valuable suggestions

Comment 1: Figure 1: if possible, improve the photographs. C, D and E are uninformative.

Response: Thank you for your critically review and for your valuable suggestions. We have added a new figure with better quality and panels C, D and E have been removed.

Comment 2: Table 1: improve the table. Maybe just lower the font size.

Response: Improved as per your  kind suggestion.

Comment 3: The caption of Figure 2 was not clear to me. It gives the impression that there are individuals with the haplotype under investigation who are affected and who are not affected. As well as affected and unaffected individuals without the haplotype. I suggest making this information clearer. If there really are affected and non-affected carriers and non-carriers, I suggest exploring this further in the paper.

Response: As per your kind suggestion, we have revised the legends to figure 2 to make it clearer.